# Chemical Characterization and Antioxidant, Antibacterial, Antiacetylcholinesterase and Antiproliferation Properties of *Salvia fruticosa* Miller Extracts

**DOI:** 10.3390/molecules28062429

**Published:** 2023-03-07

**Authors:** Michella Dawra, Jalloul Bouajila, Marc El Beyrouthy, Alain Abi Rizk, Patricia Taillandier, Nancy Nehme, Youssef El Rayess

**Affiliations:** 1Laboratoire de Génie Chimique, Université de Toulouse, CNRS, INPT, UPS, 31326 Toulouse, France; 2Faculty of Agricultural Engineering and Veterinary Medicine, Lebanese University, Dekwaneh, Beirut P.O. Box 6573, Lebanon; 3Department of Agriculture and Food Engineering, School of Engineering, Holy Spirit University of Kaslik, Jounieh BP 446, Lebanon

**Keywords:** *Salvia fruticosa* extracts, antiproliferation activity, antioxidant capacity, antibacterial activity, polyphenols

## Abstract

The *Salvia fruticosa* (Mill.) is the most medicinal plant used in Lebanon. The aim of this study is to investigate the phytochemical composition and the biological activities (in vitro) of its extracts. The plant was extracted by cold maceration with four solvents presenting an increasing polarity: cyclohexane (CHX), dichloromethane (DCM), ethyl acetate (EtOAc) and methanol (MeOH). The extracts were screened for their chemical composition by a HPLC-DAD detector for phenolic compounds identification and quantification and by GC-MS for volatile compounds detection. The antioxidant capacity (DPPH inhibition) was tested. Biological activities, mainly anti-Alzheimer activity (acetylcholinesterase inhibition), the antiproliferation of two human colon cancer cell lines (HCT-116 and Caco-2 cells) and antibacterial activity, were evaluated. Ten aromatic compounds were quantified by HPLC-DAD analysis. A total of 123 compounds were detected by GC-MS analysis. The MeOH extract showed a very interesting antioxidant activity with an inhibition percentage (IP) of 76.1% and an IC_50_ of 19.4 μg/mL. The EtOAc extract exhibited the strongest inhibition against the acetylcholinesterase activity (IP = 60.6%) at 50 μg/mL. It also strongly inhibited the proliferation of the HCT-116 cells (IP = 87.5%), whereas the DCM extract gave the best result with the Caco-2 cells (IP = 72.3%). The best antibacterial activity was obtained with the MeOH extract against *Staphylococcus aureus* (MIC = 1.2 μg/mL) and with the EtOAc extract against *Escherichia coli* (MIC = 2.4 μg/mL). This study highlights the chemical composition and therapeutic potential of *S. fruticosa*. It is important to mention that the following chemical compounds were identified for the first time in plant extracts: 2,6,11,15-tetramethyl-hexadeca-2,6,8,10,14-pentaene; 4,5,6,7-tetrahydroxy-1,8,8,9-tetramethyl-8,9-dihydrophenaleno [1,2-b]furan-3-one; podocarpa-1,8,11,13-tetraen-3-one,14-isopropyl-1,13-dimethoxy; podocarpa-8,11,13-trien-3-one,12-hydroxy-13-isopropyl-,acetate; 3′,8,8′-trimethoxy-3-piperidin-1-yl-2,2′-binaphthyl-1,1′,4,4′-tetrone; and 2,3-dehydroferruginol, thus underlining the originality of this study.

## 1. Introduction

In recent decades, due to the excessive need for healthy medications devoid of harmful synthetic and chemical products, there was a growing interest in finding new efficient, non-toxic and natural bioactive compounds. Aromatic and medicinal plants, including Salvia species, are known to have a potent role in the treatment of various illnesses, such as aches, epilepsy, colds, bronchitis, tuberculosis, hemorrhage and menstrual disorders [1]. Although there are around 900 species of *Salvia*, only a few are commercially important [2]. Particularly, the *Salvia fruticosa* Miller (*S. fruticosa* Mill.), also named *S. libanotica* and formerly *S. triloba*, belonging to the Lamiaceae family, is an endemic species of the Mediterranean Basin generally and Lebanon specifically. It is also known as the East Mediterranean sage or Lebanese sage and represents most of the imported sage in the United States rather than the *S. officinalis* [3]. It is considered the most widely used medicinal plant in Lebanon since ancient times [4] and grows at an altitude of 200–400 m [5]. In folk medicine, *S. fruticosa* aerial parts are used by herbalists and pharmacists, either internally as infusions to treat cold symptoms, mouth and throat inflammation, cough [4] and abdominal pain [6] or applied externally. The essential oils of the *Salvia fruticosa* were widely investigated and more than 100 volatile compounds were identified in several sage species. They mainly belonged to the classes of monoterpenes, sesquiterpenes, diterpenes and non-isoprenoid compounds, usually with thujone, camphor and 1,8-cineole as the most dominant ones [7]. Regarding non-volatiles, about 160 polyphenolic compounds were identified from sage plants: flavonoids and their glycosides, anthocyanins, and phenolic acids with characteristic caffeic acid derivatives, such as rosmarinic acid, phenolic diterpenes, such as carnosic acid, and other phenolic glycosides [7,8]. The latter play an important role in the prevention against oxidative damages [5,6,7,8,9]. It is very important to draw to the attention that, in the case of thujone, notwithstanding many beneficial effects, the tremendously neurotoxic danger of this volatile organic constituent has been checked out [10]. Conforming to the European Medicines Agency EMA/HMPC (2016), the daily exposure of 6.0 mg (for a 2-week duration) is allowed. Thereby, it is extremely important to control the thujone content in sage and sage-based products. Contrariwise, rosmarinic acid is one of the most useful flavonoids resulting in sage and derivative products, due to its value through anticancer and antioxidant treatments [11]. Moreover, several studies revealed that *S. fruticosa* essential oils exhibited pharmacological properties, for instance, antibacterial, antioxidant, anticholinesterase and antiproliferative activities [12]. Salvia is also known for its use as a culinary herb and an ornamental plant that has a sweet nectar and pollen for pollinators. Economically, it is easily accessible and has an affordable price in the market.

In the literature, the medicinal properties of *S. fruticosa* are often related to volatile compounds in essentials oils. However, the non-volatile compounds can also contribute to the importance of this plant. The primary purpose of this study was to carry out comprehensive research and provide data on the phytochemistry and biological activities of *S. fruticosa* extracts (CHX, DCM, EtOAc and MeOH) grown in Lebanon. The antioxidant and biological activities of the *S. fruticosa* were identified in vitro through testing the radical scavenging effects on DPPH, then the capacity of each extract to inhibit the AChE enzyme, the proliferation of HCT-116 and Caco-2 cells, and the antibacterial activity against several bacterial strains.

## 2. Results

### 2.1. Plant Materials and Extraction Yields

The yields of the four extracts of *S. fruticosa* are shown in Figure 1. The highest one was obtained with the MeOH extract (9.2%), followed by the CHX extract (3.6%), DCM extract (3.4%) and EtOAc extract (1.1%).

Dincer et al. reported that *Salvia* sp. extraction yields ranged between 17.8 and 20.3% [1]. Bozan et al. showed that the methanolic extracts yields of eight *Salvia* species including the *S. halophila* Hedge, *S. tomentosa* Miller, *S. fruticosa* Miller, *S. chrysophylla* Stapf, *S. sclarea* L, *S. clicica* Boiss. and Kotschy, *S. cryptantha* Montbret and Aucher ex Bentham, and *S. palaestina* Bentham varied between 12.8 and 26.3% [13]. The highest yield (26.3%) was recorded with the *S. fruticosa* Miller. Variations in the yields and composition could be affected by many factors, such as the plant development stage, extraction and ecological conditions, soil nature and geographical coordinates [4].

### 2.2. Total Phenolic Content

The largest amount of phenolic compounds was obtained with the MeOH extract and was 135.1 mg GAE/g of dw as shown in Figure 2. This indicates that either the majority of the phenolic compounds of the *S. fruticosa* were polar or the most abundant ones were polar. The TPCs of the other extracts were 54.4, 44.4 and 23.5 mg GAE/g of dw, respectively, for the DCM, EtOAc and CHX extracts.

Depending on the weather conditions through the years, the TPC detected in the aerial part, especially in the leaves of the *S. fruticosa*, ranged between 63.7 and 144 mg GAE/g dw [12]. Dincer et al. stated that the methanolic extracts of the *S. fruticosa* collected from Turkey presented a TPC between 41.5 and 44.6 mg GAE/g dw, almost 3.2 times lower than the amount obtained in the current study [2]. Moreover, they mentioned that the TPC of the MeOH extracts of *S. fruticosa* was more abundant than the one obtained with other *Salvia* species extracts. Duletić-Laušević et al. obtained a TPC of 132 mg GAE/g dw in the methanolic extract of the *S. fruticosa*, which is slightly lower than the one obtained in the current study [14]. Salvia with a high phenolic content has several interesting applications. It can be used in oily food because of its significant capacity to reduce undesirable fragrances, extend shelf life, delay the formation of toxic oxidation products, increase nutritional value and prevent microbial growth. It is also widely employed in the cosmetic industry [15].

### 2.3. Identification and Quantification of Phenolic Compounds by HPLC-DAD

Ten compounds, of which nine were phenolic compounds and one a methoxy-phenolic compound, were detected and quantified by HPLC-DAD as reported in Figure 3 and Table 1.

The identification of the compounds by HPLC-DAD was based on the comparison of the HPLC retention times and the DAD spectra to those found in the literature. It is worth mentioning that the detected compounds were found for the first time in *S. fruticosa* extracts. The amount of compound **10** was the highest in the CHX extract and decreased gradually in the DCM and EtOAc extracts underlining its non-polar character. Polydatin (compound **4**), which was found in the EtOAc and MeOH extracts, presented the highest amount in the MeOH extract, which underlines its polarity. These results are in agreement with those obtained with the TPC determination. The MeOH extract presented the highest TPC mainly because of the presence of polydatin, which was the most abundant phenolic compound detected by HPLC-DAD.

The majority of the compounds (70% of the detected molecules) were extracted by the non-polar solvents (CHX and DCM). The CHX extract contained the following compounds: 5′,3′-dihydroxyflavone (**5**) (y=0.0126x−0.0317; 0.09 mg/g); 3-benzyloxy-4,5-dihydroxy-benzoic acid methyl ester (**7**) (y=0.0961x+0.5481; 0.1 mg/g); pinosylvin monomethyl ether (**9**) (y=0.1265x−0.5347; 0.9 mg/g); and 3,6,3′-trimethoxyflavone (**10**) (y=0.1017x+0.1091; 0.9 mg/g). The rutin (**3**) (y=0.1029x+0.6179; 0.3 mg/g), 5,7-dihydroxy-4-phenylcoumarine (**6**) (y=0.1605x−0.0115; 1.6 mg/g) and 4′,5-dihydroxy-7-methoxyflavone (**8**) (y=0.1159x+2.1574; 0.35 mg/g) were extracted by the DCM solvent. The EtOAc extract contained the 3-amino-4-hydroxybenzoic acid (**1**) at 0.1 mg/g (y=0.5959x+0.4365), the 3,4-dihydroxy-5-methoxybenzoic acid (**2**) at 7.7 mg/g (y=0.1682x−0.047)  and the polydatin (**4**) at 2.7 mg/g (y=0.0445x−0.0083). Compound (**1**) was also found in the MeOH extract but at a higher amount (0.3 mg/g). Polydatin (**4**) was the most abundant compound and was present at 74.3 mg/g in the MeOH extract. Compound **6** was present in the MeOH extract (0.1 mg/g) at a lower amount than in the DCM extract (1.6 mg/g). Compounds **1**, **4**, **6** and **10** were detected in several extracts of *Salvia fruticosa* but at different concentrations depending on their polarity and solubility.

### 2.4. GC-MS Analysis of the S. fruticosa Extracts before and after Derivatization (Trimethylsilylation)

A total of 58 compounds were identified by GC-MS before derivatization (trimethylsilylation) and 65 additional ones after derivatization (Table 2).

A total of 68.1% of the detected compounds were present in the CHX extract underlining their non-polar nature. This is the first study that analyzes the volatile compounds of the *S. fruticosa* organic extracts. It is important to mention that this research has allowed us to reveal, for the first time, the presence of the following molecules in plants extracts: 2,6,11,15-tetramethyl-hexadeca-2,6,8,10,14-pentaene (***30***); 4,5,6,7-tetrahydroxy-1,8,8,9-tetramethyl-8,9-dihydrophenaleno[1,2-b]furan-3-one (***38***); podocarpa-1,8,11,13-tetraen-3-one, 14-isopropyl-1,13-dimethoxy- (***40***); podocarpa-8,11,13-trien-3-one, 12-hydroxy-13-isopropyl-, acetate (***41***); 3′,8,8′-trimethoxy-3-piperidin-1-yl-2,2′-binaphthyl-1,1′,4,4′-tetrone (***45***); and 2,3-dehydroferruginol (***33′***). Some molecules were found for the first time in *S. fruticosa* and were the following: 4-terpineol (***4***); humalane-1,6-dien-3-ol (***36***); (+/−)-demethylsalvicanol (***46***); 12-O-methylcarnosol (***49***); β-eudesmol (***5′***); cuminyl alcohol (***7′***); androstenediol (***35′***); kolavenol (***36′***); 6,7-dihyroferruginol (***38′***); 2-palmitoglycerol (***40′***); 2-monostearin (***48′***); monoolein (***49′***); 2-monolinolenin (***51′***); cytosine (***54′***); campesterol (***55′***); germanicol (***58***); α-amyrin (***59′***); and micromeric acid (***65′***). Some volatile compounds detected in the extracts of this study were previously found in the EO of many *Salvia* species. For instance, camphor (***12***), β-pinene (***4***) and caryophyllene oxide (***25***) were detected in the EO of *S. lavandulaefolia* [16]. Camphene (***2***), *p*-cymene (***5***), α-thujone (***9***), β-thujone (***11***), endo-borneol (***14***), terpinen-4-ol (***16***), β-caryophyllene (***19***) and viridiflorol (***26***) were found in the EO of the African *S. officinalis* [17]. Some alkanes, such as heptacosane (***51***) and octacosane (***52***), were detected in the EO of the *Salvia hierosolymitana* Boiss growing wild in Lebanon [18]. Furthermore, uvaol (***58***), betulin (***61***), oleanolic acid (***63′***) and ursolic acid (***64′***) were previously found in the 70% acetone-water extract of the *S. officinalis* and the rosmarinic acid (***60′***) was found in the *S. officinalis*, *S. limbata*, *S. virgata*, *S. hypoleuca*, *S. macrosiphon* and *S. choloroleuca* [19]. As shown in Table 2, some compounds existed in two or three extracts, such as the 2-monopalmitin (***45***) and carnosol (***45′***). This behavior is attributed to the extraction process intended to soften and break the plant’s cell walls in order to release the soluble phytochemicals. The amount of each compound will then depend on its affinity to each organic solvent, mainly on its polarity and solubility.

### 2.5. DPPH Assay for the Determination of the Antioxidant Activity

The antiradical activity of the aerial part of the *S. fruticosa* organic extracts was assessed and compared with the standard ascorbic acid (IC_50_ = 4 μg/mL). The concentration of the extracts was adjusted to 50 μg/mL. As listed in Table 3, it can be stated that the strongest inhibition of DPPH was obtained with the MeOH extract (76.1%) with an IC_50_ value of 19.4 μg/mL. It was followed by the EtOAc extract, which showed an inhibition percentage of 20.9%, then the DCM extract with 6.5% and finally no inhibition was registered with the CHX extract.

The strongest DPPH inhibition was obtained with the MeOH extract. Several authors tested the capacity of the MeOH extracts of many Libyan, Turkish and Iranian *Salvia* species to quench the DPPH free radical and found that the IC_50_ values were as follows: *S. fruticosa*, IC_50_ = 36.37 μg/mL; *S. multicaulis*, IC_50_ = 386.9 μg/mL, *S. viridis*, IC_50_ = 570 μg/mL [20]; and *S. macrosiphon*, IC_50_ = 2743.05 μg/mL [21]. Interestingly, the concentrations were, respectively, 9, 97, 142 and 686 times higher than those obtained with the MeOH extract of the current study (IC_50_ = 19.4 μg/mL). El Boukhary et al. assessed the ability of *S. fruticosa* MeOH extracts prepared from the roots and the aerial parts to inhibit DPPH. The inhibition percentages obtained were 32.16 and 41.5%, respectively, and were 2.37 and 1.83 times less effective than the MeOH extract of the current study (76.1%) [3]. Moreover, the capacity of the MeOH extract to inhibit the DPPH was 3.3 times less effective than that of the ascorbic acid (19.4 vs. 4.0 μg/mL). Nevertheless, the antioxidant activity of the methanolic extract was significant and resulted from the presence of several molecules in the extract. The fractioning and separation of these molecules may give more interesting IC_50_ values than those obtained with the ascorbic acid. The high correlation coefficient (R^2^ = 0.92) between the phenolic content of the extracts and their corresponding antioxidant activity confirmed that the phenolic compounds were the main components responsible for the antioxidant activity of the *S. fruticosa* extracts. The MeOH extract, which exhibited the highest antioxidant activity, contained, per gram of extract, 0.3 mg of 3-amino-4-hydroxybenzoic acid; 74.3 mg of polydatin; and 0.1 mg of 5,7-dihydroxy-4-phenylcoumarine (Table 1). It also contained salvianolic acid A (***22′***); 2,3-dehydroferruginol (***33′***); and rosmarinic acid (***60′***) (Table 2). These molecules may have worked synergistically in order to give a good inhibition. In the EtOAc extract, the presence of phenol, 2,2′-methylenebis 6-(1,1-dimethylethyl)-4-methyl-(***43***); 4-hydroxybenzoic acid (***9′***); and 2-palmitoyl glycerol (***40′***) that have a labile proton may have contributed to the quenching of the DPPH free radical (Table 2). The CHX and DCM extracts showed a poor antioxidant activity. However, some of their chemical compounds may have inhibited the DPPH free radical, such as the β-sitosterol (IC_50_ = 140 μg/mL) [22], caryophillene oxide (IC_50_ = 84.0 μg/mL [23]), carnosol (IC_50_ = 0.59 μM [24]) and carnosic acid (IC_50_ = 60 μM) [24]. The lupeol (***56***) was previously shown to exhibit a good antioxidant behavior. In fact, it quenched the DPPH free radical by 50.0% at 70 μg/mL [25].

### 2.6. Biological Activities

#### 2.6.1. Antiacetylcholinesterase Activity (Anti-AChE)

The analysis was carried out with 50 μg/mL of each *S. fruticosa* extract. The results were compared to that of the standard GaHbr. As shown in Table 3, a similar inhibition percentage was registered for the CHX (59.5%), DCM (60.5%) and EtOAc (60.6%) extracts. Although the MeOH extract presented the lowest inhibition with 52.46%, it was not significantly different from the previous ones (*p* ˃ 0.05). This behavior against the AChE enzyme could be attributed to many molecules that are present in the samples. Ayaz et al. proved that the β-sitosterol exhibited a considerable AChE inhibition with an IC_50_ value of 55 μg/mL. This compound was present in the non-polar CHX and DCM extracts (***55***-Table 2) [22]. Savelev et al. isolated the (−)-β-pinene (***4***-Table 2), the caryophyllene oxide (***26***-Table 2) and the (−)-camphor (***12***-Table 2) from *S. lavandulaefolia* EO [16]. These compounds were also found in the *S. fruticosa* CHX and/or DCM extracts. The (−)-β-pinene inhibited the AChE activity with an IC_50_ value of 0.2 mg/mL, while the camphor reduced its activity by 39.0% at 0.5 mg/mL. Recently, Karakaya et al. proved that the caryophyllene oxide isolated from *Salvia verticillata* subsp. Amasiaca EO presented a good anti-AChE potency since it reduced the enzyme activity by 41.4% at 200 μg/mL. It can be concluded from the previous data that the anti-AChE activity of the extracts resulted from complex interactions, both synergistic and antagonistic, between their terpene constituents [23].

#### 2.6.2. MTT Assay for the Measurement of the Antiproliferation Activity

The antiproliferation activity of the *S. fruticosa* extracts prepared at 50 μg/mL was tested on two lines of cancer cells: Caco-2 and HCT-116. The tamoxifen was used as a positive control. The highest growth inhibition of the Caco-2 cells was registered with the DCM extract (72.3%), followed by the EtOAc extract (62.1%), whereas the highest inhibition of the HCT-116 cells was recorded with the EtOAc extract (87.5%), followed by the DCM extract (70.7%) (Table 3). The CHX extract was not active at all, while the MeOH extract only reduced the growth of the HCT-116 cells by 7.2% (*p ≤* 0.05). Therefore, the *S. fruticosa* extracts were more active against the proliferation of the HCT-116 cells than the Caco-2 ones. The American National Cancer Institute (NCI) considered that the extracts with an IC_50_ < 30 μg/mL had a promising cytotoxic activity (Suffness, 1990), which is the case of the DCM and EtOAc extracts of the current study (Table 3) [25]. Duletić-Laušević et al. tested the antiproliferation activity of the Libyan *S. fruticosa-* EtOH extract against the HCT-116 cells and found that it was able to reduce their growth with an IC_50_ of 375.96 μg/mL [14]. The obtained value was 25.7 times higher than the one recorded with the EtOAc extract (14.6 μg/mL) of the current study. The latter gave the best antiproliferation activity and showed a promising anticancer application. Polydatin was found to inhibit the growth of Caco-2 cells and gave an IC_50_ of 74.9 μg/mL [26]. This compound was present in the EtOAc extract that presented an IC_50_ of 31.1 μg/mL when tested against the same cancer cell line. Therefore, polydatin may have contributed to the antiproliferation activity of the EtOAc extract along with other compounds leading to a better result than when tested alone.

#### 2.6.3. Antimicrobial Activity Assay

The four *S. fruticosa* extracts were tested individually for their capacity to inhibit the growth of seven foodborne pathogenic bacterial strains including four Gram-negative and three Gram-positive bacteria. The minimum inhibitory concentrations (MICs) obtained are displayed in Table 4.

The DCM extract significantly inhibited the growth of *S*. Kentucky (MIC = 19.5 μg/mL) and moderately that of *L. monocytogenes* ATCC 19115 (MIC = 78.1 μg/mL). The MeOH extract showed an important antibacterial activity against *S. aureus* ATCC 25923 and *E. coli* ATCC 8739 with very low MIC values of 1.2 and 2.4 μg/mL, respectively. The antibacterial activity of the EtOAc extract against *E. coli* was the same as for the MeOH one (same MIC value of 2.4 μg/mL). The EtOAc extract also showed a moderate activity against *L. monocytogenes* ATCC 19115 (MIC = 39 μg/mL). Duletić-Laušević et al. tested the antimicrobial activity of a Libyan *S. fruticosa* extract (EtOH and water) against *E. coli*, *S.* Enteritidis, *S. aureus* and *L. monocytogenes* and found MIC values of 1500, 1500, 1000 and 1000 μg/mL, respectively [14]. When comparing these values to the best MIC values obtained in the current study against the same bacterial species, we find that the MIC values of this study were, respectively, 625, 19.21, 833 and 208 times lower than those found in the literature, underlining the important antibacterial activity of our extracts. However, we should take into consideration that the extraction solvents used, the geographical origin and the bacterial strains were not the same. Based on the study conducted by Kosová et al., the EtOAc extract’s behavior could be explained by the presence of the 4-hydroxybenzoic acid (***9′***-Table 2) considered a preservative [27]. This acid was shown to inhibit the growth of *S. aureus* and *E. coli* at 20 mmol/l. The extracts of the current study exhibited a bacteriostatic activity against some of the bacterial strains tested, while some others, mainly Gram + ones, were not affected. This can be explained by the presence of resistance mechanisms in these strains [28].

### 2.7. Principal Component Analysis (PCA)

The PCA analysis was used in this study in order to establish the relation between the different biological activities and the chemical composition of the extracts. As shown in Table 5, the axes of inertia have been hidden from this analysis.

The percentage of total variation was recorded at 98.6% and proven by the structuring accessions in Figure 4.

The axes were retained because they expressed 57.1% (PC1) and 41.5% (PC2). Simultaneously, the loadings in the PCA loading plots expressed how good the correlation was between the major components and the original variables studied. There was a very good correlation between the antioxidant activity and the TPC. PC1 was highly correlated only with the TPC with a loading of 0.93 (Table 6).

The second axis was well correlated with HCT-116, Caco-2 and AChE with loadings of 0.73, 0.63 and 0.59, respectively (Table 6). When applying the principal component analysis, it seemed that there was a discriminate structure. The oval forms grouped the different extracts into three classes: C1 (*S. fruticosa*—DCM and *S. fruticosa*—EtOAc), C2 (*S. fruticosa*—CHX) and C3 (*S*. *fruticosa*—MeOH). Since the two plots (biplot) were gathered together, it can be noticed that the high TPC and antioxidant activity were related to the *S. fruticosa*—MeOH extract. In addition, the *S. fruticosa*—DCM, *S. fruticosa*—EtOAc and *S. fruticosa*—CHX (poor in TPC) were located symmetrically in the negative side of the PC1 axis, which suggested that the high antiproliferation and antiacetylcholinesterase activities of these extracts were not only related to the phenolic compounds.

## 3. Materials and Methods

### 3.1. Chemicals and Plant Materials

In October 2018, the fresh leaves of the *S. fruticosa* were collected from Naher Ibrahim, Lebanon. The botanical identification was made by Dr. Marc El Beyrouthy, chairman and general manager of the company Nature by Marc Beyrouthy. The samples were deposited in the Herbarium of Botany, Medicinal Plants and Malherbology, School of Engineering, Holy Spirit University of Kaslik, Lebanon under the registry number MNIIIb177a. The analytical standards used for the identification and quantification of the main phenolic compounds found in the plant extracts were as follows: 3-amino-4-hydroxybenzoic acid; 3,4-dihydroxy-5-methoxybenzoic acid; rutin; polydatin; 5′,3′-dihydroxyflavone; 5,7-dihydroxy-4-phenylcoumarine; 3-benzyloxy-4,5-dihydroxy-benzoic acid methyl ester; 4′,5-dihydroxy-7-methoxyflavone; pinosylvin monomethyl ether; and 3, 6,3′-trimethoxyflavone, all of which were obtained from Sigma-Aldrich (St Louis, MO, USA).

### 3.2. Preparation of the Extracts

The collected aerial parts of *S. fruticosa* were dried in shade at an ambient temperature and transformed into powder with a particle size of 0.8 mm [29]. A cold maceration with four solvents presenting an increasing polarity (CHX, DCM, EtOAc and MeOH) was performed to yield the four organic extracts. A total of 100 g of powder was successively extracted with 2 L of each solvent during 2 h with an agitation of 300 rpm. The filtrates were recovered after filtration through whatman filter papers (Fisher, Asiane, France). The extracts were obtained by evaporating the solvent under vacuum at 35 °C. The extraction yield was calculated as follows: $Yield\;\left(\%\right)=\left(\frac{m}{M}\right)\times 100$, “*m*” being the dry weight obtained in grams and “*M*” being the weight of the plant material in grams.

### 3.3. Total Phenolic Content Determination

The total phenolic content (TPC) of each extract was evaluated spectrophotometrically at 765 nm using the Folin–Ciocalteu (FC) method as described by Dawra et al. [29]. Gallic acid (0–115 μg/mL) was used for the calibration curve. The results were expressed as mg of gallic acid equivalents (GAE)/g dw.

### 3.4. HPLC-DAD Fingerprint

The HPLC analysis was performed in an ultimate 3000 pump—Dionex and Thermo separation product detector DAD model (Thermo Fisher Scientific, Waltham, MA, USA). Separation was achieved on an RPC18 reversed-phase column (Phenomenex, Le Pecq, France), 25 cm × 4.6 mm and particle size of 5 μm, thermostated at 25 °C as described by Dawra et al., with modifications [28]. The elution was performed at a flow rate of 1.2 mL/min, using a mobile phase consisting of MilliQ water (pH 2.6) (solvent A) and acidified water/MeCN (20:80 *v*/*v*) (solvent B). The samples were eluted by the following linear gradient: from 12% B to 30% B for 35 min, from 30% B to 50% for 5 min, from 50% B to 88% B for 5 min and finally from 88% B to 12% B for 15 min. The extracts were prepared at a concentration of 20 mg/mL using the mixture acidified water/MeCN (80:20 *v*/*v*) and then filtered through a Millex-HA 0.45 µm syringe filter (Sigma Aldrich). Then, 20 μL of each sample were injected and the detection was registered at 280 nm. The phenolic compounds were identified by comparison to the retention time of some known standards and then quantified using their corresponding calibration curves.

### 3.5. Gas Chromatography GC-MS Analysis

The identification of the volatile compounds of the organic extracts, before and after derivatization, was conducted using the protocol described by Dawra et al., with some modifications [29]. The analyses were conducted using an Agilent gas chromatograph 6890 coupled to a 5975 Mass Detector. The 7683 B auto sampler injected 1 μL of each extract. A fused silica capillary column DB-5 MS (30 m × 0.25 mm internal diameter, film thickness 0.25 μm) (Supelco, Sigma-Aldrich, Darmastadt, Germany) was employed. The temperature ramp was settled between 35 and 300 °C. The column temperature was initially set to 35 °C before being gradually increased to 85 °C at 15 °C/min, held for 20 min at 85 °C, raised to 300 °C at 10 °C/min and finally held for 5 min at 300 °C. Helium (purity 99.99%) was used as a carrier gas at a flow rate of 0.8 mL/min. Mass spectra were registered at 70 eV with an ion source temperature held at 310 °C and a transfer line heated at 320 °C. The record of each acquisition was made in full-scan mode (50–400 amu). The main target was to find the maximum resemblance in terms of spectra between the compounds found in the extracts and those suggested by the NIST08 database (National Institute of Standards and Technology, https://www.nist.gov/ (accessed on 15 April 2021)), using AMDIS software, and the retention time was used to facilitate many tasks. The derivatization method consisted of dissolving 5 mg of each extract in 1 mL of its own solvent except for the MeOH extract. The latter was dissolved in MeCN. After that, 150 μL of BSTFA and 1.5 μL of TMSC were added to the solution. The mixture was agitated for 30 s in order to increase the solubility. The reaction mixture was maintained at 40 °C for 30 min. A total of 10 μL of each derivatized solution were injected into the GC-MS and analyzed as previously reported.

### 3.6. Free Radical Scavenging Activity: DPPH Test

The antioxidant scavenging activity of the extracts was examined using the DPPH method as described by Dawra et al. [29]. A total of 20 μL of the diluted plant extract (500 μg/mL) was added to 180 μL of a 0.2 mM methanolic DPPH solution in a 96-well microplate (Micro Well, Thermo Fisher Scientific, Bordeaux, France). After an incubation period of 30 min at 25 °C, the absorbance was measured at 515 nm. The antioxidant activity was expressed as the inhibition percentage of DPPH using the following equation: $\%\;\mathrm{INB}=100\times \left(\frac{A_{\mathrm{blank}}-A_{\mathrm{sample}}}{A_{\mathrm{blank}}}\right)$. The extract concentration providing a 50% reduction in the DPPH initial absorbance (IC_50_) was calculated using the linear relation between the extract concentrations and the corresponding % INB of DPPH. All measurements were performed in quadruplicate.

### 3.7. Biological Activities

#### 3.7.1. Antiacetylcholinesterase Activity

The antiacetylcholinesterase (AChE) activity was tested using the Ellman’s procedure as previously reported by Dawra et al. [29]. In a 96-well microplate, 50 μL of 0.1 mM sodium phosphate buffer (pH = 7.5), 125 μL of DTNB (5,5′-dithiobis-2-nitrobenzoic acid), 25 μL of the diluted plant extract (500 μg/mL) and 25 μL of the enzyme solution (493.2 U) were mixed. The microplate was incubated at 25 °C for 15 min. Then, 25 μL of ACTHI was added and the final blend was incubated at 25 °C for 25 min. Finally, the absorbance was measured at 421 nm. The *A_blank_* was measured without the extract. The inhibition percentage of the enzyme activity was calculated as follows: $\%\;\mathrm{INB}=100\times \left(\frac{A_{\mathrm{blank}}-A_{\mathrm{sample}}}{A_{\mathrm{blank}}}\right)$.

#### 3.7.2. Antiproliferation Activity

The antiproliferation activity of the plant extracts was assessed on two different types of human colon cancer cells (HCT-116 and Caco-2). The test was based on the MTT reduction by the mitochondrial dehydrogenases of intact cells to a purple formazan product. MTT is a yellow water-soluble tetrazolium salt. The cell lines were purchased from Sigma-Aldrich (Manassas, VA, USA). A volume of 100 μL of a suitable culture medium containing 3 × 10^4^ cells was added to each well of a 96-well microplate. Then, 100 μL of the same culture medium containing the plant extract was added. The final concentration of the extract in each well was 50 μg/mL. The culture media used were, respectively, the RPMI 1640 (Sigma Aldrich, St. Louis, MO, USA) for the HCT-116 cells and the Dulbecco’s modified Eagle’s medium GlutaMAX (DMEM, Sigma Aldrich, USA) for the Caco-2 cells. The microplate was incubated at 37 °C for 48 h. The supernatant was then removed, and 50 μL of the MTT solution was added followed by an incubation of 40 min. After removing the MTT reagent, 80 μL of DMSO was added to solubilize the formazan crystals. Finally, the absorbance was measured at 605 nm. The tamoxifen was used as a positive control whereas the negative control was composed of the cell suspension without the plant extracts (blank). The inhibition percentage of the cells’ proliferation was calculated as follows: $\%\;\mathrm{INB}=100\times \left(\frac{A_{\mathrm{blank}}-A_{\mathrm{sample}}}{A_{\mathrm{blank}}}\right)$.

#### 3.7.3. Antimicrobial Activity Assay

The Gram-negative strains used in this assay were *Escherichia coli* ATCC 8739 and the Kentucky, Infantis and Enteritidis serotypes of *Salmonella enterica* provided by the Lebanese Agriculture Research Institute (LARI), Lebanon. The *Salmonella* serotypes were isolated from chicken samples collected from slaughterhouses. The Gram-positive strains were *Staphylococcus aureus* ATCC 25923, *Listeria monocytogenes* ATCC 19115 and *Listeria monocytogenes* isolated from “fish-filet” at the LARI. A bacterial suspension of each bacterial strain was prepared in a Mueller–Hinton Broth (MHB) at a concentration of 2 × 10^8^ CFU/mL (0.5 McFarland standard) [30]. The minimum inhibitory concentration (MIC) values of the *Salvia fruticosa* extracts were determined by serial dilution in a 96-well microplate. Each well first contained 100 μL of MHB. The dried extracts were dissolved in pure DMSO to a concentration of 5 mg/mL. Then, the extract solutions were half-diluted with MHB to obtain a concentration of 2.5 mg/mL. Afterwards, 100 μL of the latter were placed in the first well and a serial dilution was conducted in order to obtain the following concentrations in each row: 1250, 625, 312.5, 156.2, 78.1, 39, 19.5, 9.7, 4.8, 2.4 and 1.2 μg/mL. Next, 100 μL of each bacterial strain tested were added to the extract solutions. Therefore, the initial bacterial concentration was adjusted to 10^8^ CFU/mL in each well. The negative control was composed of 100 μL of DMSO and 100 μL of the bacterial strain tested while the positive control contained 100 μL of MHB and 100 μL of the bacterial strain. The absorbance was measured at time 0 min and after an overnight incubation at 37 °C (24 h) using a Multiskan Sky Microplate Spectrophotometer (Thermo Fisher Scientific, Cleveland, OH, USA).

### 3.8. Statistical Analysis

The data represent the mean of four replicates ± standard deviation (SD). The results were subjected to a multiway analysis of variance, and the mean comparisons were performed by a Tukey’s multiple range test using SPSS version 20.0 (Statistical Package for the Social Sciences, Inc., Chicago, IL, USA). The differences between means were considered significant at *p*-value < 0.05. The linear correlation coefficient (R^2^) was calculated to establish the relationship between the TPC and the antioxidant or any other biological activity. For exploratory data analysis, the results were processed by one of the multivariate analysis techniques, the principal components analysis (PCA). The PCA was conducted using XLSTAT (version 2020.1, Addinsoft, Pearson edition, Waltman, MA, USA) for a better discrimination between the studied parameters.

## 4. Conclusions

This novel research shed light on the phytochemistry and biological activities of the *Salvia fruticosa* Miller extracts native to Lebanon. The HPLC-DAD analysis permitted the identification and quantification of ten aromatic compounds; nine of which were recognized as phenolic compounds. The GC-MS analysis revealed the presence of 123 volatile compounds. *S. fruticosa* extracts showed interesting biological activities. The MeOH extract showed a high antioxidant activity. The four extracts presented a good antiacetylcholinesterase activity. The DCM and EtOAc extracts revealed a significant antiproliferation activity against the HCT-116 and Caco-2 cancer cells. Interestingly, the four extracts exhibited an excellent antibacterial activity against pathogenic foodborne Gram-negative and Gram-positive bacteria with low MIC values, particularly against *Escherichia coli*, *Staphylococcus aureus* and *Listeria monocytogenes.* The above-mentioned promising pharmacological activities highlight the plant’s potential use in the development of new antimicrobial drugs. They encourage us to identify and purify the bioactive compounds by performing a bioguided fractionation of the most active extracts. The extracts that possess interesting antioxidant and antimicrobial activities should also be tested in food preservation. In vivo studies of the most active compounds should also be performed in order to better assess their therapeutic potential.

## Figures and Tables

**Figure 1 molecules-28-02429-f001:**
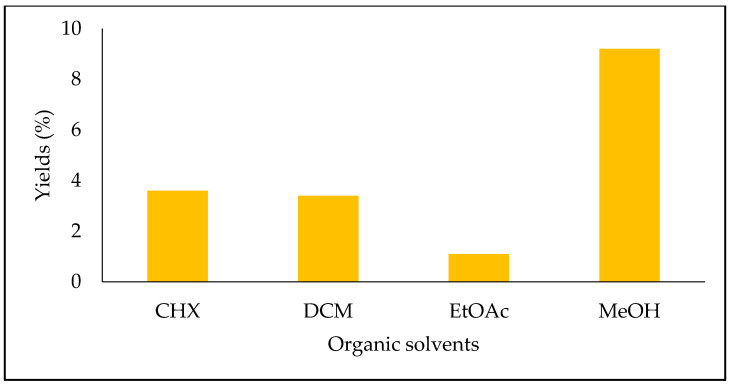
Extraction yields of *Salvia fruticosa* extracts (%).

**Figure 2 molecules-28-02429-f002:**
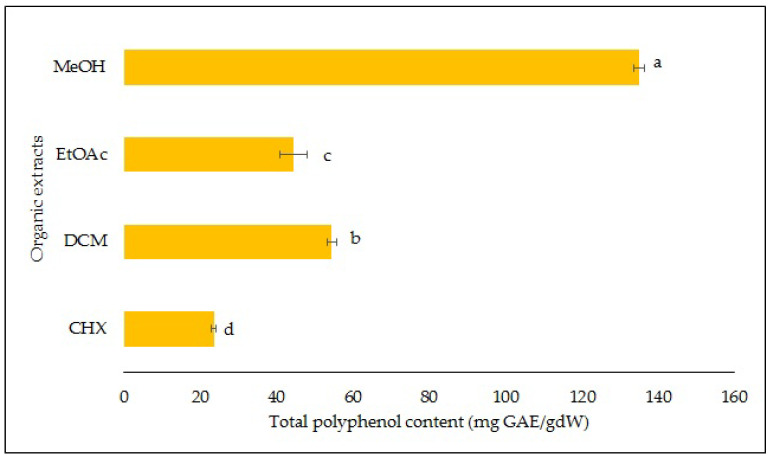
Total polyphenol content. Results are means ± standard deviations of four independent analyses. Different letters indicate significantly different means at *p* < 0.05.

**Figure 3 molecules-28-02429-f003:**
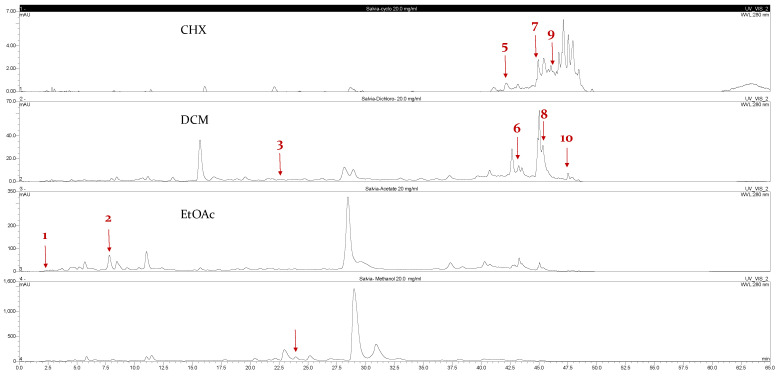
HPLC chromatograms of the four extracts of S. fruticosa: CHX, DCM, EtOAc and MeOH extracts. 3-amino-4-hydroxybenzoic acid (**1**); 3,4-dihydroxy-5-methoxybenzoic acid (**2**); rutin (**3**); polydatin (**4**); 5′,3′-dihydroxyflavone (**5**); 5,7-dihydroxy-4-phenylcoumarine (**6**); 3-benzyloxy-4,5-dihydroxy-benzoic acid methyl ester (**7**); 4′,5-dihydroxy-7-methoxyflavone (**8**); pinosylvin monomethyl ether (**9**); 3,6,3′- trimethoxyflavone (**10**).

**Figure 4 molecules-28-02429-f004:**
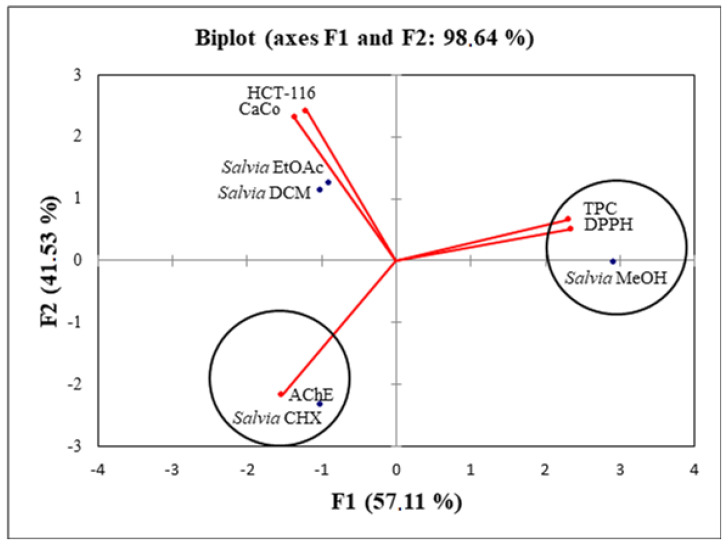
Principal component analysis biplot of the antioxidant and biological activities of the *S. fruticosa* extracts.

**Table 1 molecules-28-02429-t001:** Identification and quantification of phenolic compounds in *S. fruticosa* extracts by HPLC-DAD.

				*S. fruticosa* Extracts (mg of Compound/g of Extract)
N°	RT (min)	Compounds	Chemical Structure	Calibration Curves	CHX	DCM	EtOAc	MeOH
**1**	2.2	3-amino-4-hydroxybenzoic acid	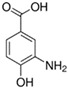	y=0.5959x+0.4365			0.1 ± 0.0	0.3 ± 0.0
**2**	7.7	3,4-dihydroxy-5-methoxybenzoic acid	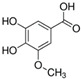	y=0.1682x−0.047			7.7 ± 0.1	
**3**	22.6	Rutin	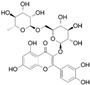	y=0.1029x+0.6179		0.3 ± 0.1		
**4**	23.3	Polydatin	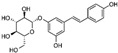	y=0.0445x−0.0083			2.7 ± 1.2	74.3 ± 0.0
**5**	42.1	5′,3′-dihydroxyflavone	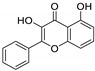	y=0.0126x−0.0317	0.1 ± 0.0			
**6**	43.4	5,7-dihydroxy-4-phenylcoumarine	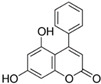	y=0.1605x−0.0115		1.6 ± 0.1		0.1 ± 0.0
**7**	44.9	3-benzyloxy-4,5-dihydroxy—benzoic acid methyl ester	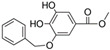	y=0.0961x+0.5481	0.1 ± 0.0			
**8**	46.2	4′,5-dihydroxy-7-methoxyflavone	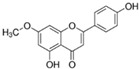	y=0.1159x+2.1574		0.3 ± 0.0		
**9**	47.0	Pinosylvin monomethyl ether	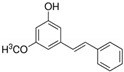	y=0.1265x−0.5347	0.9 ± 0.0			
**10**	47.9	3, 6,3′-trimethoxyflavone	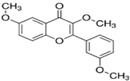	y=0.1017x+0.1091	0.9 ± 0.1	0.6 ± 0.0	0.2 ± 0.0	

The results are the means of duplicate experiments (±SD).

**Table 2 molecules-28-02429-t002:** Compounds detected by GC-MS in the four extracts of *S. fruticosa* before and after derivatization (trimethylsilylation).

N°	RI	Compounds	CHX	DCM	EtOAc	MeOH
** *1* **	-	α-thujene	+			
** *2* **	-	camphene	+			
** *3* **	-	sabinene	+			
** *4* **	-	(−)-β-pinene	+			
** *5* **	908	*p*-cymene	+			
** *6* **	911	eucalyptol	+	+		
** *7* **	962	cis-4-thujanol	+			
** *8* **	1105	sabinene hydrate	+			
** *9* **	1110	thujone	+			
** *10* **	1114	α-monoacetin		+		
** *11* **	1118	β-thujone	+			
** *12* **	1140	(−)-camphor		+		
** *13* **	1141	(+)-2-bornanone	+			
** *14* **	1164	endo-borneol	+			
** *15* **	1166	δ-terpineol	+			
** *16* **	1178	4-terpineol	+			
** *17* **	1196	α-terpineol	+			
** *18* **	1276	α-terpinyl acetate	+			
** *19* **	1290	caryophyllene	+			
** *20* **	1292	aromadendrene	+			
** *21* **	1294	humulene	+			
** *22* **	1298	γ-muurolene	+			
** *23* **	1527	1,5,9-trimethyl-1,5,9-cyclododecatriene	+			
** *24* **	1546	espatulenol	+			
** *25* **	1549	caryophyllene oxide	+	+		
** *26* **	1554	(+)-viridiflorol	+	+		
** *27* **	1563	humulene oxide II	+			
** *28* **	1575	caryophylladienol II	+			
** *29* **	1576	cubenol	+			
** *30* **	1745	2,6,11,15-tetramethyl-hexadeca-2,6,8,10,14-pentaene	+			
** *31* **	1850	neophytadiene		+	+	
** *32* **	1931	palmitic acid, methyl ester				++
** *33* **	1937	7,9-di-tert-butyl-1-oxaspiro(4,5)deca-6,9-diene-2,8-dione	+	+		
** *34* **	1977	palmitic acid	+			
** *35* **	2080	epimanool	+	+		
** *36* **	2112	humulane-1,6-dien-3-ol	+			
** *37* **	2131	linoleic acid	+			+
** *38* **	2318	4,5,6,7-tetrahydroxy-1,8,8,9-tetramethyl-8,9-dihydrophenaleno[1,2-b]furan-3-one	+	+	+	
** *39* **	2358	totarol	+			
** *40* **	2370	podocarpa-1,8,11,13-tetraen-3-one, 14-isopropyl-1,13-dimethoxy-	+	+	+	
** *41* **	2406	podocarpa-8,11,13-trien-3-one, 12-hydroxy-13-isopropyl-, acetate		+	+	
** *42* **	2421	pregnan-3-yl acetate		+		
** *43* **	2445	phenol, 2,2′-methylenebis[6-(1,1-dimethylethyl)-4-methyl-	+	+	+++	
** *44* **	2517	2-monopalmitin	+	+		+++
** *45* **	2571	3′,8,8′-trimethoxy-3-piperidin-1-yl-2,2′-binaphthyl-1,1′,4,4′-tetrone		+		
** *46* **	2633	(±)-demethylsalvicanol	+	+		
** *47* **	2638	retinoic acid	+			
** *48* **	2738	α-monostearin		+		
** *49* **	2822	12-O-methylcarnosol		+		
** *50* **	2901	α-tocospiro B	+			
** *51* **	2903	heptacosane	+	+		
** *52* **	2950	octacosane	+	+		
** *53* **	2963	vitamin E acetate	+	+		
** *54* **	3006	tetratetracontane	+	+		
** *55* **	3065	β-sitosterol	+	+		
** *56* **	-	lupeol	+			
** *57* **	-	ursolic aldehyde		+		
** *58* **	-	uvaol	+	+		
**After Derivatization**						
**N°**	RT(min)	Compounds	CHX	DCM	EtOAc	MeCN
** *1′* **	9.04	2,3-butanediol		+		
** *2′* **	9.8	carbamic acid	+			
** *3′* **	10.5	1-butoxy-2-propanol	+			
** *4′* **	17.4	exo-borneol	+			
** *5′* **	24.5	β-eudesmol	+			
** *6′* **	25.1	glycerol	+			
** *7′* **	33.3	cuminyl alcohol	+			
** *8′* **	36.0	linolool oxide	+	+		
** *9′* **	38.15	4-hydroxybenzoic acid			+	
** *10′* **	38.17	24-epicampesterol	+			
** *11′* **	38.3	spathulenol	++			
** *12′* **	38.9	4-tert-butylcatechol		+		
** *13′* **	39.0	glutaric acid				+
** *14′* **	40.6	L-(−)-sorbose				+++
** *15′* **	40.7	α-talofuranose				+++
** *16′* **	40.9	(Z)-5,8,11-eicosatrienoic acid		+		
** *17′* **	41.0	methyl α-D-glucofuranoside				+
** *18′* **	41.5	β-D-(+)-talopyranose				++
** *19′* **	42.4	D-mannopyranose				+
** *20′* **	42.48	methyl caffeate		+		
** *21′* **	42.6	D-ribofuranose				+
** *22′* **	43.1	salvianolic acid A				+
** *23′* **	43.2	ferulic acid			+	
** *24′* **	43.38	palmitelaidic acid	+			
** *25′* **	43.47	myo-inositol				+
** *26′* **	43.49	scyllo-inositol				+
** *27′* **	43.67	(13S)-labda-8(20),14-dien-13-ol	+	+++	+++	
** *28′* **	43.68	caffeic acid	+	+		
** *29′* **	43.9	phytol	+	+	+	
** *30′* **	44.0	t-cadinol	+	+	+	
** *31′* **	44.3	α-linolenic acid	+++			
** *32′* **	44.5	stearic acid	+++			
** *33′* **	44.6	2,3-dehydroferruginol	+	+	+	+
** *34′* **	44.7	ferruginol	++	+		
** *35′* **	45.1	androstenediol		+		
** *36′* **	45.46	kolavenol	+			
** *37′* **	45.49	sempervirol	+			
** *38′* **	47.0	6,7-didehydroferruginol		++		
** *39′* **	47.05	rosmadial	+	++		
** *40′* **	47.1	2-palmitoylglycerol			+	
** *41′* **	47.2	D-(+)-turanose				+++
** *42′* **	47.3	1-monopalmitin	+		+	+++
** *43′* **	47.4	carnosic acid		++		
** *44′* **	47.7	lactulose				+
** *45′* **	47.9	carnosol	+	+++	+	
** *46′* **	48.1	sucrose				+
** *47′* **	48.2	D-trehalose				+++
** *48′* **	48.5	2-monostearin			+	
** *49′* **	48.61	monoolein	+			
** *50′* **	48.62	rosmanol		+		
** *51′* **	48.67	2-monolinolenin	++			
** *52′* **	49.0	ethinyl estradiol	+			
** *53′* **	51.2	α-tocopherol	+	+		
** *54′* **	51.9	cytosine		+		
** *55′* **	52.2	campesterol	+			
** *56′* **	52.4	stigmasterol	+			
** *57′* **	53.1	β-amyrin	+			
** *58′* **	53.2	germanicol	+			
** *59′* **	53.5	α-amyrin	+			
** *60′* **	53.9	rosmarinic acid				+
** *61′* **	54.7	betulin	+			
** *62′* **	54.79	erythrodiol	+	+		
** *63′* **	55.7	oleanolic acid	+++	+++	+++	
** *64′* **	56.3	ursolic acid	+++	+++	+	
** *65′* **	57.1	micromeric acid		+		

Note: +++—high presence; ++—average presence; +—low presence; RI is the retention index determined in regards to a series of n-alkanes (C_7_–C_35_) on the apolar DB-5 MS. The RI values are identified as following: $RI=100x\;n+100x\left(\left(\frac{\mathrm{Apex}\;\mathrm{Rt}\;\left(\mathrm{lower}\;\mathrm{alkane}\right)-\mathrm{Rt}\;\left(\mathrm{lower}\;\mathrm{alkane}\right)}{\mathrm{Rt}\;\left(\mathrm{follower}\;\mathrm{alkane}\right)-\mathrm{Rt}\;\left(\mathrm{lower}\;\mathrm{alkane}\right)}\right)\right)\;$ where *n* is the number of carbon of lower alkane.

**Table 3 molecules-28-02429-t003:** Antioxidant and biological activities of *S. fruticosa* extracts tested at 50 µg/mL.

Extracts and Standards	DPPH (% INB)	IC_50_ (µg/mL)	Anti-AChE (% INB)	HCT-116 Cells (% INB)	IC_50_ (µg/mL)	Caco-2 Cells (% INB)	IC_50_ (µg/mL)
CHX	na	˃50	59.5 ± 1.5 ^a^	na	˃50	na	˃50
DCM	6.5 ± 2.5 ^c^	˃50	60.5 ± 5.0 ^a^	70.7 ± 3.6 ^b^	19.7 ± 3.6	72.3 ± 3.4 ^a^	24.0 ± 3.4
EtOAc	20.9 ± 15.5 ^b^	˃50	60.6 ± 4.3 ^a^	87.5 ± 6.1 ^a^	14.6 ± 6.1	62.1 ± 0.3 ^b^	31.1 ± 0.3
MeOH	76.1 ± 1.2 ^a^	19.4 ± 3.2	52.4 ± 1.7 ^a^	7.2 ± 2.1 ^c^	˃50	na	˃50
Ascorbic acid	80.0 ± 12.6	4.0 ± 0.1	-	-	-	-	-
GaHBr	-	-	88.1 ± 0.5	-	-	-	-
Tamoxifen	-	-	-	91.1 ± 1.0	-	87.3 ± 0.1	-

Notes: na—not active. Results are the means (±SD) of four independent analyses. Different letters means significant differences between the values obtained with the extracts of the same organ according to Tukey’s test (*p* ≤ 0.05).

**Table 4 molecules-28-02429-t004:** Minimum inhibitory concentration (MIC) values obtained with the *S. fruticosa* extracts tested against seven foodborne Gram-positive and Gram-negative pathogenic bacteria.

Bacterial Strains	MIC (μg/mL)				
		CHX	DCM	EtOAc	MeOH
*Salmonella* Enteritidis	Gram- bacteria	78.1 ± 1.2 ^c^	312.5 ± 0 ^c^	-	625 ± 0 ^d^
*Salmonella* Kentucky		625 ± 1.2 ^d^	19.5 ± 0 ^a^	-	-
*Salmonella* Infantis		39 ± 1.5 ^b^	625 ± 1.39 ^d^	-	78.1 ^c^
*Escherichia coli* ATCC 8739		625 ± 1.2 ^d^	312.5 ± 1.2 ^c^	2.4 ± 0.4 ^a^	2.4 ± 0.21 ^b^
*Listeria monocytogenes* ATCC 19115	Gram + bacteria	4.8 ± 4.9 ^a^	78.1 ± 0 ^b^	39 ± 0 ^b^	-
*Listeria monocytogenes* Fish filet		-	-	-	-
*Staphylococcus aureus* ATCC 25923		-	-	-	1.2 ± 0 ^a^

“-”: no inhibition. Results are the means (±SD) of two independent analyses. Different letters mean significant differences between the values obtained with the same extract tested on different bacterial strains according to Tukey’s test (*p* ≤ 0.05).

**Table 5 molecules-28-02429-t005:** Contribution of variable factors to the principal component analysis (%).

	F1	F2
TPC	32.6	2.6
DPPH	33.2	1.5
AChE	14.2	28.5
HCT-116	8.7	35.1
Caco-2	11.1	32.1

**Table 6 molecules-28-02429-t006:** Correlation established between variables and factors.

	F1	F2
TPC	**0.93**	0.05
DPPH	**0.94**	0.03
AChE	0.40	**0.59**
HCT-116	0.24	**0.73**
Caco-2	0.31	**0.66**

## Data Availability

Not applicable.

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
