# Peer review of "Chemical Characterization and Antioxidant, Antibacterial, Antiacetylcholinesterase and Antiproliferation Properties of Salvia fruticosa Miller Extracts"

_molecules, 2023, doi:10.3390/molecules28062429_

Round 1
Reviewer 1 Report
Dear Authors,
The MS entitled " Chemical characterization, antioxidant, antibacterial, anti-acetylcholinesterase and anti-proliferation properties of Salvia fruticosa Miller extracts" was thoroughly reviewed. The MS describes the biological activities of various types of extarcat from a medicinal plant, Salvia fruticosa. The study of extracts is often not sufficient and needs to isolate the compounds responsible for such activities however, the authors have characterized the extracts and also justified their potency. The study is satisfactory and a minor revision is needed. My comments are:
1. Write the composition of major compounds in abstract.
2. Describe the local name of plant in introduction.
2. Line 317. What is meant by 119 Sigma-Aldrich (USA)?
3. Line 321: CHX stands for?
4. Write all the equations separately and in proper format using Microsoft word equation tool. Also number them.
5. line 387. What is ACTHI? Also write DTNB in full.
6. line 391-408. Correct font.
7. Line 441. Conclusion should be revised and the names of compounds should be omitted rather they should be placed in the abstract.
8. the references should be updated as these are older.
9. Formatting of the tables is necessary.
Author Response
Dear Reviewer
Thank you for your comments
Please see the answer in the attachment

Reviewer 2 Report
The manuscript of M. Dawra, et al., “Chemical characterization, antioxidant, antibacterial, anti-acetylcholinesterase and antiproliferation properties of Salvia fruticosa Miller extracts” describes compositions of four (hexane, dichloromethane, ethyl acetate and methanol) successive extracts of samples of Greek sage harvested in Lebanon. Antioxidant, antibacterial, anti-acetylcholinesterase and antiproliferation properties of these extracts were studied.
Reviewer’s notes.
General notes. 1. Compare. Lines 308—309: “The botanical identification was made by Dr. Marc EL BEYROUTHY (written by capital letters, Rev.), chairman and general manager of the company Nature by Marc Beyrouthy“. Simultaneously, Dr. Marc Beyrouthy is a co-author of this study (line 5). In Author Contributions: “funding acquisition, M.E.B. and J.B” (lines 468—469), and in Acknowledgements: “…and Nature by Marc Beyrouthy for their technical and financial support” (line 475), contributions of Dr. Beyrouthy are stated. However, “no external funding” is declared (line 470). One can conclude a hidden conflict of interests, which is in a contradiction with the statement in line 476: “Conflicts of Interest: The authors declare no conflict of interest”. So, is the last statement true? If not, conflict of interests must be declared.
2. List of references is too scarce for this extensively studied subject. Some relevant publications (which are not cited in the manuscript) are given below. Of course, this list cannot be considered as excessive.
1. M.D. Gkoni, K. Zeliou, V.D. Dimaki, P. Trigas, and F.N. Lamari, GC-MS and LC-DAD-MS Phytochemical Profiling for Characterization of Three Native Salvia Taxa from Eastern Mediterranean with Antiglycation Properties. Molecules 2023, 28(1), 93; https://doi.org/10.3390/molecules28010093 and refs. therein.
2. N. Pachura, A. Zimmer, K. Grzywna, A. Figiel, A. Szumny, and J. Lyczko, Chemical investigation on Salvia officinalis L. Affected by multiple drying techniques – The comprehensive analytical approach (HS-SPME, GC–MS, LC-MS/MS, GC-O and NMR). Food Chem., 2022, 397, 133802. https://doi.org/10.1016/j.foodchem.2022.133802 and refs. therein.
3. S. Süzgeç-Selçuk, T. Özek, G. Özek, S. Yur, F. Göger, M. B. Gürdal, G. G. Toplan, A. H. Meriçli, and K. H. Can Başer, The Leaf and the Gall Volatiles of Salvia fruticosa Miller from Turkey: Chemical Composition and Biological Activities. Rec. Nat. Prod., 2021, 15 (1) 10—24.
4. C. Formisano, F. Senatore, N. Apostolides Arnold, F. Piozzi and S. Rosselli, GC and GC/MS Analysis of the Essential Oil of Salvia hierosolymitana Boiss. Growing Wild in Lebanon. Natl. Prod. Commun., 2007, 2 (2), 181—184 and refs. therein.
5. D. Pitarokili, O. Tzakou, A. Loukis, C. Harvala, Volatile metabolites from Salvia fruticosa as antifungal agents in soilborne pathogens. J. Agric. Food Chem., 2003, 51(11), 3294—301. doi: 10.1021/jf0211534.
3. To improve readability, a list of abbreviations would be useful.
4. Supplementary Materials addendum is strongly recommended to support validity of identifications.
Other notes.
1. Line 17, Abstract. “CHXhexane (CHX)”: is it cyclohexane or hexane? Please, specify.
2. Lines 70—72. “Results are means ± standard deviations…”: standard deviations are missed.
3. Line 99 and below, 2.3. Identification and quantification of phenolic compounds by HPLC-DAD.
a. Identification of these compounds using r.t. and UV spectra obtained by DAD looks unreliable. One cannot estimate reliability of identification because there are no UV spectra available. ESI-MS/MS spectra would be more informative.
b. There are no peaks related to compound 1 in all four chromatograms (Figure 3): why this compound is claimed as “identified” (lines 118 and 133)?
c. There are large peaks at 15 min (DCM), 29 min (DCM, EtOAc and MeOH), 31 min (MeOH) and 42 min (DCM). The assignments are not done.
This part of the study cannot be considered as complete and reliable.
4. Line 134 and below, 2.4. GC-MS analysis of the S. fruticose extracts before and after derivatization.
a. Table 2 (lines 137—138). One might be surprised: what derivatization was performed? The answer can be found much later (lines 366—367): this is trimethylsilylation.
b. Same. For intact compounds, retention indices (RI) are presented. For TMS derivatives, retention times are given. Why is so different?
c. Table 2, item 23. 3,5-di-tert-butylphenol is not a natural product, this is a contaminant. It definitely has an antioxidant activity and can corrupt the corresponding assay if really present.
d. Table 2, item 24. “1,5,9-trimethyl-1,5,9-CHXdodecatriene” CHX is definitely excessive.
e. Table 2, item 46. This is not a natural product (contaminant, plasticizer).
f. Table 2, item 47. Same (synthetic contaminant, also line 148). Same 1’, 2’ and 3’.
g. Table 2, item 17’. Analytical artifact (methylation during extraction).
h. Table 2, item 24’. Wrong formula. Same for 8’(corrupted under copy&paste procedure?) and 32’.
i. Table 2, item 42’. Apparently, contaminant (cannot be derivatized by TMSCl).
k. Table 2, item 54’. This is not a natural product (steroidal component of contraceptive pills).
l. Table 2, items 46’ and 47’. Due to small difference in r.t., this is expectedly the same compounds (TMS derivative of sucrose).
5. Line 150: “aromadandrene” should be aromadendrene. It was found previously (see Pachura 2022, Gkioni 2023, Suzgec-Selcuk 2021, etc.). Same for γ-muurolene. Acclaimed priorities must be checked carefully before resubmission.
6. Table 2 and discussion below. Glycerol monoesters are likely to be contaminants (components of detergents for labware washing) than natural components. So, priority of their finding is questionable.
7. Line 222. Ref. [14] is missed.
8. Line 271, missed ref. [16] (see line 515, Extra “1”: to be corrected).
Author Response
Dear Reviewer
Thank you for your comments
Please see the answers in the attachment

Reviewer 3 Report
Dear Authors,
The manuscript proposed for publication is of interest in the field and in addition brings more originality due to the compounds that were identified for the first time in S. fruticosa. The manuscript is well structured and presented, it includes a series of analyzes and data that confirm the importance of this plant.
Please find bellow some recommendations:
1. Section 2.1. The data (letters) are not visible on the graph.
2. Section 2.2. : - To correct the number of the figure in the text. Figure 2 instead of 1.
- Could it be possible that the higher results obtained for TPC be due to the fact that they were determined on all aerial parts and the other authors reported only on leaves?
3. Section 2.3.: Line 209- I suggest specifying a literature data that confirms that these are really the compounds reported for the first time in S. fruticosa, considering that they were not identified by HPLC- DAD-ESI-MS method.
- Are the 10 compounds the only ones identified and quantified or were they presented only because of their originality? Data from the literature regarding the other phenolic compounds present in S. fruticosa would be indicated.
Line 111 - it is specified that compound 10 is found in the largest amount in the CHX extract, but this is not highlighted in the chromatogram.
- Is there any study to confirm that polydatin is mainly composed in S. fruticosa or is it also found in other species of sage?
Line 133: ...the compounds 1,4,6 and 10 were detected in several extracts - literature data?
Section 2.4.: Line 135- Sixty compounds were identified by GC-MAS before derivatization and 67 additional ones after derivatization- to correct the title of the table in which only the 67 after derivatization are listed.
Section 2.5.: Line 199 - are there data in the literature to confirm this? The same question for the extract in ETOAc.
Conclusions: Due to the large volume of data and analyzes performed, it would be of interest to emphasize the results obtained more clearly and in detail. For example, the antioxidant and antiproliferative activities are included in a single sentence each...
There are few literature sources, taking into account the diversity of the data presented in the study.
I can suggest some bibliographic references:
- https://www.scielo.br/j/cta/a/YxPKBR6nkk7vqQ6njBtjMBh/?format=pdf&lang=en
Author Response

(The authors gave the same response as above.)

Round 2
Reviewer 2 Report
Reviewer’s notes.
General notes. 1. … If not, conflict of interests must be declared.
Response: … However, in order to avoid ambiguity, the name of the company was removed from the acknowledgments section.
Ans. Problem is solved
2. List of references is too scarce for this extensively studied subject. Some relevant publications (which are not cited in the manuscript) are given below. Of course, this list cannot be considered as excessive.
……………….
Response: please be informed that the suggested articles were screened for their relevance and whether their content enhances the value of the article in question. Accordingly, some of them were added.
Ans. Response is accepted.
3. To improve readability, a list of abbreviations would be useful.
Response: a list of abbreviations is provided with the corrected version of the manuscript.
Ans. Sorry, the list of abbreviations is missed from the second submission.
4. Supplementary Materials addendum is strongly recommended to support validity of identifications.
Response: a supplementary materials file that contains the identification of the RI is provided.
Ans. This is a useful update, however, it would be better to GC-MS TIC chromatograms with captions.
Other notes.
1.
Line 17, Abstract.
“CHXhexane (CHX)”: is it cyclohexane
or hexane? Please, specify.
Response: it is cyclohexane
and it was corrected as requested.
Ans. Response is
accepted.
2. Lines 70—72. “Results are means ± standard deviations…”: standard deviations are missed.
Response: the sentence was adjusted upon request.
Ans. Response is accepted.
3. Line 99 and below, 2.3. Identification and quantification of phenolic compounds by HPLC- DAD.
a. Identification of these compounds using r.t. and UV spectra obtained by DAD looks unreliable. One cannot estimate reliability of identification because there are no UV spectra available. ESI-MS/MS spectra would be more informative.
Response: we are aware that the use of LC coupled to MS is more interesting for quantification. However, the calibration curves of the standards used were displayed in table 1 at λ= 280 nm. Every analyte was expressed as mg of standards/g of extract. Then, it is important to highlight that one of our main goals was the quantification of phenolic compounds. Therefore, the use of HPLC-DAD method is considered sufficient and meets all the needed analytical requirements. Moreover, providing the UV spectra would be useful when testing the validation of new analytical methods, which is not the case in this study.
Ans. a1. UV spectra are continuous (in contrast to MS which are discrete), and, hence, less reliable for identification than MS, especially if there is a substantial overlap of chromatographic peaks (for example, CHX, 42 —49 min). How deconvolution has been done? One can be in firm confidence taking absorbance at definite wavelength / appropriate r.t. that this absorbance is related to the desired compound, not the different one with the same or close r.t., or front/tail of the nearest (overlapping) chromatographic peak. Again, the use of LC-ESI-MS/MS is not “more interesting”, but much more reliable for complex mixtures of a priori unknown composition which the studied extracts are.
a2. Still no UV spectra available (can be placed in Supporting Materials: HPLC experiment vs. reference spectrum).
b. There are no peaks related to compound 1 in all four chromatograms (Figure 3): why this compound
is claimed as “identified” (lines 118 and 133)?
Response: compound 1 which is the -3-amino-4-hydroxybenzoic acid was identified and quantified in the EtOAc and MeOH extracts
only, as listed in table 1. The area of each peak follows the beer- lambert equation A = εlc. If the molecule has a significant concentration with a very weak epsilon value, the peak intensity/surface will not be clearly marked, which is the case of compound 1. Moreover,
the measure scale displayed
on the Y axis is expressed in mAu. The four chromatograms are presented
in this way in order to show the majority of the peaks obtained regardless
if we could identify them all or not. The quantification was done based on the establishment
of the calibration curve of
each standard.
Ans. The peaks arrowed as 1 are negligible for both EtOAc and MeOH extracts. “If the molecule has a significant concentration with a very weak epsilon value, the peak intensity/surface will not be clearly marked…” — sure, what about different (smaller?) wavelength? The reliability of identification is still questionable (see a2).
c. There are large peaks at 15 min (DCM), 29 min (DCM, EtOAc and MeOH), 31 min (MeOH) and 42 min (DCM). The assignments are not done. This part of the study cannot be considered as complete and reliable.
Response: some peaks were not assigned for the simple reason that we didn’t have in our research institution, the adequate standards corresponding to their retention times.
That is not a good answer: one can find reference samples using cooperation or use another approach.
However, the identification of 9 phenolic compounds and one methoxy phenol for the first time in these salvia extracts is a significant finding.
Hopefully yes, if it would be reliable.
Reply cannot be considered as satisfactory. The reviewer’s note “This part of the study cannot be considered as complete and reliable.” remains valid.
4. Line 134 and below, 2.4. GC-MS analysis of the S. fruticosa extracts before and after derivatization.
a. Table 2 (lines 137—138). One might be surprised: what derivatization was performed? The answer can be found much later (lines 366—367): this is trimethylsilylation.
Response: “trimethylsilylation” was added to the titles of section 2.4 and table 2.
Ans. Response is accepted.
b. Same. For intact compounds, retention indices (RI) are presented. For TMS derivatives, retention times are given. Why is so different?
Response: we are aiming to be always credible, that is why the identification of the compounds before derivatization was made by placing each molecule between the alkane that precedes it and the one that follows it. Thus, the RI will be more reliable especially when its formula is taking into consideration the presence of these two alkanes. Therefore, the ambiguity that may result from the identification by the RT method can be easily avoided. The formula of the RI was added as footnotes to table 2.
For TMS derivatives, the identification of the volatile compounds were done by analogy with our publication accepted in flavor and fragrance journal (DOI: 10.1002/ffj.3646), that is why we didn’t calculate the RI.
Ans. Response is accepted.
c. Table 2, item 23. 3,5-di-tert-butylphenol is not a natural product, this is a contaminant. It definitely has an antioxidant activity and can corrupt the corresponding assay if really present.
Response: thank you for drawing our attention to this point. First of all, Zhao et al. (2020) (doi: 10.3390/toxins12010035), discussed the natural source of 2,4-Di-tert-Butylphenol and its analogs including the compound in question (3,5-di-tert-butylphenol). Second, the fact that this contaminant is present depends widely on the nature of the soil in which the plant species has grown. Third, since it was detected it should be mentioned.
Ans. Sure, and marked as an artifact. Anyway, have you negative control experiment? Expectedly, no.
Finally, we are in the first phase of the study that is a chemical screening, thus we must certainly
expect to have a very complex matrix. When the purification process will be done through bioguided fractionation which is the
next step of this work, the natural
active compounds will be well isolated.
Ans. Thank you for interesting reference (Zhao, 2020). On the other hand, I
would like to recommend you several reviews and discussion papers on analytical
artifacts; Middleditch & Zlatkis, J. Chromatogr. Sci., 1987, 25, 547;
Zenkevich, DOI:
10.1134/S1061934813130108; B.O.Keller, et al., doi:10.1016/j.aca.2008.04.043
(including Supporting Materials); J.S. McIndoe, DOI: 10.1016/j.ijms.2019.04.001 and tutorial (auth. by Dr. W.W.
Christie), https://www.lipidmaps.org/resources/lipidweb/lipidweb_html/ms/others/msartefacts/index.htm
d. Table 2, item 24. “1,5,9-trimethyl-1,5,9-CHXdodecatriene” CHX is definitely excessive. Response: the compound name was corrected as requested.
Ans.
Response is accepted.
e. Table 2, item 46. This is not a natural product (contaminant, plasticizer).
Response: in the review paper, “Phthalic Acid Esters: Natural Sources and Biological Activities” (https://doi.org/10.3390/toxins13070495), Huang et al. (2021), proved that the PAEs including phthalic acid, decyl octyl ester (compound 46 of table 2), can be easily isolated and purified from various algae, bacteria, fungi and other natural sources.
Ans. Thank you for the reference. Again: found during GC(HPLC)-MS analysis does not mean present in the analyte. Again: have you negative control (blank experiment)? Both reviews in Toxins are in fact lists of papers where tert-alkyl phenols / phthalates have been mentioned. To collect does not mean to prove.
f. Table 2, item 47. Same (synthetic contaminant, also line 148). Same 1’, 2’ and 3’.
Response: Salem et al. (2016) (http://dx.doi.org/10.1016/j.bse.2016.02.024), found the 3',8,8'- trimethoxy-3-piperidin-1-yl-2,2'-binaphthyl-1,1',4,4'-tetrone (compound 47 of table 2), in the leaves of Boscia angustifolia A. Rich. (Capparaceae). Therefore, it is not surprising to find it in our extracts.
Ans. See above.
g. Table 2, item 17’. Analytical artifact (methylation during extraction).
Response: Tomala et al. (2022) (https://doi.org/10.56499/jppres22.1342_10.3.551), found the same compound in the butanolic fraction of the aqueous extract of Malva sylvestris leaves.
h. Table 2, item 24’. Wrong formula. Same for 8’ (corrupted under copy&paste procedure?) and 32’.
Response: we totally agree. However, the chemical structures were completely removed from table 2 for all molecules (table formatting for more clarity and simplicity).
Ans. Response is accepted.
i. Table 2, item 42’. Apparently, contaminant (cannot be derivatized by TMSCl).
Response: the chemical structure of the 1-monopalmitin contains two-hydroxyl groups that can be derivatized by the TMSCl.
Ans. Exactly. But methyl palmitate is pictured. No hydroxyl groups.
j. Table 2, item 54’. This is not a natural product (steroidal component of contraceptive pills).
Response: in the article “Phytochemicals Targeting Estrogen Receptors: Beneficial Rather Than Adverse Effects?” (https://doi.org/10.3390/ijms18071381), Lecomte et al. (2017), reported that steroidal components including ethinyl estradiol can be found in plant extracts.
Citation. “Many natural and synthetic chemicals in the environment and in food have been reported with hormonal activity, particularly showing estrogenic potency [22]. … Bisphenol A, nonylphenol and ethinyl estradiol were also reported to be among the major environmental estrogens.” Hence, the review mentions synthetic environmental pollutants (bisphenol A, nonylphenol and ethinyl estradiol) having estrogenic activity, not natural metabolites.
l. Table 2, items 46’ and 47’. Due to small difference in r.t., this is expectedly the same compound (TMS derivative of sucrose).
Response: thank you for drawing our attention to this point. The item 46’ was removed upon your suggestion.
Ans. Response is accepted.
5. Line 150: “aromadandrene”
should be aromadendrene. It was found previously (see Pachura
2022, Gkioni 2023, Suzgec-Selcuk 2021, etc.).
Same for γ-muurolene. Acclaimed priorities
must be checked
carefully before resubmission.
Response: these two molecules were removed
from the list of new findings
in S. fruticosa
extracts.
Ans. Response is accepted.
6. Table 2 and discussion below. Glycerol monoesters are likely to be contaminants (components of detergents for labware washing) than natural components. So, priority of their finding is questionable.
Response: in the review paper “A Comprehensive Review on Phytochemistry and Pharmacological Activities of Clinacanthus nutans (Burm.f.) Lindau.” (doi: 10.1155/2018/9276260), Khoo et al. (2018), mentioned that the glycerol monoester was previously identified in the extracts of this plant species.
Ans. Please, find something more reliable than this review where trimethylsilyl derivatives are listed as “phyto constituents” (among others, including phosphonic acid, levoglucosan, phthalates/phthalic acid, tert-butyl phenols, and even “9-Azabicyclo (6.1.0) non-4-4en-9-amine”). Not every database hit appeared after clicking experimental mass spectrum results in right answer.
7.
Line 222. Ref. [14]
is missed. Response: corrections
are done.
Ans. Response is
accepted.
8. Line 271, missed ref. [16] (see line 515, Extra “1”: to be corrected). Response: corrections are done.
Ans.
Response is accepted.
New reviewer’s notes.
1. Line 297. 2.6.3. Antimicrobial activity assay. What antibiotic was used for a positive control?
2. Line 421. Injection volume 10 mkL looks too large for GC. Is it really applied? (1 mkL is mentioned above, line 405).
3. Lines 565, 585, References. Extra symbol —

Author Response
Dear Reviewer 2,
Kindly find in attachment the answer for the remaining comments
